# The Analysis of the Editing Defects in the *dyw2* Mutant Provides New Clues for the Prediction of RNA Targets of Arabidopsis E+-Class PPR Proteins

**DOI:** 10.3390/plants9020280

**Published:** 2020-02-21

**Authors:** Bastien Malbert, Matthias Burger, Mauricio Lopez-Obando, Kevin Baudry, Alexandra Launay-Avon, Barbara Härtel, Daniil Verbitskiy, Anja Jörg, Richard Berthomé, Claire Lurin, Mizuki Takenaka, Etienne Delannoy

**Affiliations:** 1Institute of Plant Sciences Paris-Saclay (IPS2), Université Paris-Saclay, CNRS, INRAE, Univ Evry, 91405 Orsay, France; bastien.malbert@ips2.universite-paris-saclay.fr (B.M.); mauricio.obando@vedascii.org (M.L.-O.); kevin.baudry@ips2.universite-paris-saclay.fr (K.B.); alexandra.launay-avon@inrae.fr (A.L.-A.); etienne.delannoy@inrae.fr (E.D.); richard.berthome@inrae.fr (R.B.); 2Institute of Plant Sciences Paris-Saclay (IPS2), Université de Paris, CNRS, INRAE, 91405 Orsay, France; 3Molekulare Botanik, Universität Ulm, 89069 Ulm, Germany; matthias.burger@uni-ulm.de (M.B.); mail@barbara-haertel.de (B.H.); daniil.verbitskiy@me.com (D.V.); mizuki.takenaka@pmg.bot.kyoto-u.ac.jp (M.T.); 4Department of Botany, Graduate School of Science, Kyoto University, Oiwake-cho, Sakyo-ku, Kyoto 606-8502, Japan

**Keywords:** *Arabidopsis*, genome containing organelles, mutants, pentatricopeptide repeat (PPR) proteins, RNA editing

## Abstract

C to U editing is one of the post-transcriptional steps which are required for the proper expression of chloroplast and mitochondrial genes in plants. It depends on several proteins acting together which include the PLS-class pentatricopeptide repeat proteins (PPR). DYW2 was recently shown to be required for the editing of many sites in both organelles. In particular almost all the sites associated with the E+ subfamily of PPR proteins are depending on DYW2, suggesting that DYW2 is required for the function of E+-type PPR proteins. Here we strengthened this link by identifying 16 major editing sites controlled by 3 PPR proteins: OTP90, a DYW-type PPR and PGN and MEF37, 2 E+-type PPR proteins. A re-analysis of the DYW2 editotype showed that the 49 sites known to be associated with the 18 characterized E+-type PPR proteins all depend on DYW2. Considering only the 288 DYW2-dependent editing sites as potential E+-type PPR sites, instead of the 795 known editing sites, improves the performances of binding predictions systems based on the PPR code for E+-type PPR proteins. However, it does not compensate for poor binding predictions.

## 1. Introduction

The bacterial ancestry of plastids and mitochondria can be seen in their gene expression systems which rely on numerous post-transcriptional regulatory steps [1]. Among these, RNA editing converts selected cytidine (C) to uridine (U) at the RNA level [2,3]. In terrestrial plants, hundreds or thousands of C-to-U editing sites have been identified in plastids and mitochondria [4,5,6]. The biological significance of this mechanism and of these sites is still not entirely understood but the associated molecular mechanisms have been widely studied in the last 30 years (for a review see [7]). The editing of each of these sites requires the recognition of its upstream cis-elements by a specific pentatricopeptide repeat (PPR) protein [8]. PPR proteins are characterized by a tandem array of degenerate 35 amino acid motifs [9] and are almost exclusively targeted to organelles [10]. The PPR proteins are sequence specific RNA binding proteins through the interaction of each PPR motif with one RNA base [11]. The specificity is driven by hydrogen bonds between two amino acids of the PPR motif and the RNA base [12]. For editing, the PPR proteins are binding upstream of the site they are associated to in a way that the last C-terminal PPR motif binds the base at position −4 [13]. Using the list of editing sites experimentally associated to characterized editing PPR proteins, a PPR RNA recognition code was discovered, refined and experimentally validated [14,15,16]. It can be used to predict the binding sites of uncharacterized PPR proteins [17]. Besides PPR proteins, several other proteins have been shown to be involved in organellar RNA editing suggesting the existence of protein complexes which were named editosomes [7]. In particular, members of a small family called Multiple Organellar RNA editing Factors (MORF/RIP) were found to be involved in RNA editing [18,19,20]. These proteins function as general editing factors, can interact with many PPR proteins [20,21] and were shown to increase PPR affinity for their target RNA [22]. 

The PPR proteins can be divided into two classes: the P-type PPR proteins and the PLS-type PPR proteins which include long (L) and short (S) variant of the canonical PPR motif [23,24]. P-type PPR proteins have been associated with a wide range of organellar RNA metabolisms [25] while PLS-type PPR proteins are almost only involved in C-to-U editing [26]. Most PLS-type PPR proteins also harbor conserved C-terminal domains: E1, E2, E+ and DYW which are used to classified PLS-type PPR proteins into corresponding subgroups [24]. While the function of the E1 and E2 domains remains unclear, the DYW domain shows strong similarities to cytidine deaminases [27]. Targeted mutagenesis experiments [28,29] as well as phylogenetic correlations [27] and the observation of C-to-U editing in *Escherichia coli* expressing a plant DYW-type PPR protein [30] strongly support that the DYW domain provides the cytidine deaminase activity required for the PPR editing factors carrying it. However, many PPR proteins involved in editing are truncated and lack a DYW domain. We and others demonstrated that they can recruit DYW domains in *trans* from members of a small PPR protein subgroup, the DYW1-like proteins [31,32]. These six proteins, which were named after DYW1 the first one to be characterized [31], carry very few PPR domains, no canonical E2 and E+ domains but end with a DYW domain [6]. In particular, DYW2 was shown to be associated with the editing of hundreds of sites in *Arabidopsis thaliana* [6]. The DYW2 dependent editing sites show a strong positive bias to editing sites associated to a particular class of PLS-type PPR proteins, the E+ PPR proteins. These PPR proteins harbor the E1, E2 and E+ domains but lack the DYW domain. The functional association of DYW2 and E+ PPR proteins as well as their physical interactions both *in vitro* and *in vivo* [6,32] suggest that the DYW domain is provided by DYW2 exclusively to E+ PPR proteins.

In this study, we strengthened the link between E+ PPR proteins and DYW2 in *A. thaliana* by the refined analysis the *dyw2* mutant and the identification of 16 editing sites associated to 3 PLS-type PPR proteins, PGN, MEF37 and OTP90. We used this functional association to improve the binding predictions of the *A. thaliana* E+ PPR proteins.

## 2. Results

### 2.1. The Editing Defects in the dyw2 Mutant Define 2 Classes of RNA Editing Targets

The RNA-seq analysis of the *dyw2-1* KO mutant identified 392 differentially edited sites compared to the wild type (WT) and 261 were considered to depend on DYW2 [6]. This analysis covered 95 sites associated with known PPR proteins and showed that none of the 47 analyzed sites associated with DYW-PPR proteins depend on DYW2 whereas all of the 35 analyzed sites associated with E+-PPR proteins depend on DYW2. It supported the requirement of DYW2 specifically for the function of E+ PPR proteins. However, 14 editing sites associated with PPR proteins were missing from this study. Slight modifications of our bio-informatic pipeline (see Material and Methods) allowed the study of all 109 editing sites associated with PPR proteins using the Guillaumot et al. dataset [6]. This re-analysis identified 529 differentially edited sites in *dyw2-1* compared to the WT (Appendix A). Among them, 487 were found in the mitochondrial genome and 42 in the chloroplastic genome. It should be noted that no genuine editing modification was found in nuclear encoded transcripts demonstrating the specificity of our improved pipeline (not shown). Among these 529 sites, all 39 sites associated with E+ PPR proteins showed a decrease of editing above 30% (*ccmB_428*, M18007 associated to CMW1 being the less affected with -31.4% of editing extent variation). Overall, 272 sites are differentially edited in *dyw2-1* with a decrease of editing extent above 25%. They don’t include any of the 55 editing sites associated with DYW-PPR proteins but one site associated to MEF19 [33], an E-PPR (*ccmB_566*, M17869) and one site associated to PPME [34], a P-PPR (*nad1_898*, M234052) (Figure 1, Appendix A). 

To further study the specificity of DYW2 regarding PPR sub-families, we extended the repertoire of characterized editing sites by studying three editing PPR proteins: PGN [35], AT5G08305 and AT1G25360. 

### 2.2. Characterization of AT1G56570 (PGN), AT5G08305 (MEF37) and AT1G25360 (OTP90) 

PGN (AT1G56570) belongs to the E+ PPR protein subfamily with 13 predicted PPR motifs upstream of E1, E2 and E+ domains and is targeted to the mitochondria [35]. Similarly, AT5G08305 encodes an E+ PPR protein with 12 identified PPR motifs followed by E1, E2 and E+ domains. It is predicted to be targeted to mitochondria (Suba4 [36]) and was named Mitochondrial Editing Factor 37 (MEF37) (Figure 2A).

The PPR protein encoded by AT1G25360 was named ORGANELLAR TRANSCRIPT PROCESSING 90 (OTP90). Containing 15 identified PPR domains upstream of E1, E2 and DYW C-terminal domains (Figure 2A), OTP90 belongs to the PLS-DYW subgroup of the PPR family [23]. OTP90 was shown to be targeted to both organelles after expression of fusions of its N-terminal 100aa peptide as well as the full length protein to the fluorescent DsRed2 protein in transient expression assays [10]. To complete these data, we confirmed the subcellular localization of OTP90 in stable transgenic *Arabidopsis* lines (Appendix A). We also showed that OTP90 interacts with MORF proteins (Appendix A). 

To decipher their molecular function, we identified mutants with T-DNA insertions in each of these three genes. The Salk_141937 line (*pgn*) was used by Laluk et al. [35] to characterize the function of PGN. We confirmed that *pgn* mutants did not show any major macroscopic phenotype in normal growth conditions (Figure 2B) but are slightly more sensitive to *Botrytis cinerea* infection (Appendix A). An RNA-seq comparison of *pgn* leaves with Col-0 ones in normal growth conditions confirmed the absence of full length *PGN* transcripts in *pgn* and identified 98 differentially expressed genes (DEG; Data S2) but no significant gene ontology enrichment. 

Two independent mutant lines were used to characterize the function of MEF37, Salk_058773 (*mef37-1*) and Salk_110420 (*mef37-2*). The analysis of the T-DNA flanking sequences identified an insertion 373 bp and 586bp from the initiation codon respectively. The *mef37-1* mutant did not show any major macroscopic phenotype in normal growth conditions (Figure 2B). Its RNA-seq analysis confirmed the absence of full length *MEF37* transcripts in leaves and identified 34 DEG (Data S2) showing a significant enrichment for genes with the GO term “Response to acid chemical”.

The analysis of the T-DNA flanking sequences in the SM_3_29659 line (*otp90*) identified an insertion 806 bp after the initiation codon. The *otp90* mutant did not show any major macroscopic phenotype in normal growth conditions (Figure 2B) but we observed shorter primary roots in *in vitro*-germinated *otp90* mutants and smaller seeds were collected on adult plants (Appendix A). These phenotypes were not observed in OTP90 complemented *otp90* lines (Appendix A). The RNA-seq analysis of *otp90* mutant confirmed the absence of full length *OTP90* transcripts in *otp90* leaves and identified 12 DEG (Data S2) including two out of the three copper/zinc superoxide dismutases (*CSD1* and *CDS2*) which are downregulated in *otp90*.

### 2.3. pgn, mef37and otp90 Mutants Are Impaired in Mitochondrial C to U Editing

PGN, MEF37 and OTP90 are proteins belonging to PPR subfamilies involved in RNA editing. We characterized the editotype of *pgn*, *mef37-1* and *otp90* by RNA-seq. Using the same analysis pipeline as for *dyw2*, we were able to quantify the editing rate at 204 to 453 editing sites in plastids and 1305 to 1441 editing sites in mitochondria. Among them, we identified several differentially edited sites in each mutant line (Table 1). 

In *mef37-1*, 11 mitochondrial sites were found differentially edited. The edition of 10 sites was inhibited from 51 to 100%. Seven of them (*ccmB_566, ccmB_551, rps3_1470, mttb_387, ccmC_179, nad4L_trailer_72* and *atp4_138*) being edited with a rate of less than 7% in the mutant when they were 70-99% edited in wild type leaves. These sites include M17869 (*ccmB_566*) which is known to be edited by MEF19. In contrast, editing of *nad4_437* was decreased by only 50% suggesting that MEF37 is not required for this site and that another editing factor is also able to target this site. Interestingly, in addition to these sites with decreased editing in the *mef37-1* mutant, the edition of M362349 (*atp4_144*) was shown to be strongly increased from 7.7% to 27.3% in *mef37-1*. Except M49473 (*atp6_71*) which was not detected in the *dyw2* analysis because its editing is hardly detectable, all the editing defects of *mef37-1,* including the increase at *atp4_144*, were shown to be similarly impacted by the *dyw2* mutation (Table 1). 

Only two mitochondrial editing sites were shown to be affected in *pgn* mutants. Editing of M8348 (*cox2_742*) and M165765 (*nad6_leader_-73*) was completely abolished with −100% of ΔEE. These 2 sites are similarly affected in the *dyw2* mutant with 11% and 3% editing, respectively (Table 1).

Finally, the detailed editotype analysis of the *otp90* mutant by RNA-seq detected 12 mitochondrial sites differentially edited in *otp90*. The edition of seven of them was strongly inhibited (from 72 to 100%). These sites include six sites highly edited in Col-0: M59321 (*nad1_500*), M18355 (*ccmB_80*), M308481 (*ccmC_184*), M219668 (*mttB_97*), M17839 (*ccmB_596*) and M191687 (*ccmFc_1246*). On the other hand, similarly to what was observed for the *atp4_144* site in *mef37-1*, the edition of four sites was increased (from 7 to 149%). None of the six major editing defects of *otp90* mutant were observed in *dyw2*. 

To confirm the molecular function of the three PPR proteins, the corresponding mutants were complemented by overexpression of ORFs under the 35S promotor (MEF37 and PGN) or expression of the complete genomic locus (OTP90). As described previously, the OTP90 complemented *otp90* lines did not show the root and seed phenotypes observed in the mutants (Appendix A). The editotypes of the three mutants and their corresponding complemented lines were analyzed by direct cDNA sequencing of the major editing sites identified during the RNAseq analysis (Table 2, Appendix A). 

All major RNA editing defects observed in RNAseq experiments were confirmed by Sanger sequencing and peak height analysis (Table 2). All these defects (8 in *mef37-2*, 2 in *pgn* and 6 in *otp90*) were restored in the corresponding complemented lines (Table 2). In particular, 6, 2 and 4 editing sites were abolished in *mef37*, *pgn* and *otp90* mutants, respectively.

### 2.4. Using the dyw2 Editing Defects to Refine the PPR Code for PLS PPR Proteins

With the molecular characterization of OTP90, MEF37 and PGN, 60 of the 205 PLS-type PPR proteins encoded in the genome of *Arabidopsis thaliana* have now been characterized and associated to 132 major editing sites both in plastids and mitochondria. In particular, 18 of the 59 E+ PPR proteins have now been associated to 49 major editing sites and all of them are DYW2-dependent reinforcing a model where the E+ PPR proteins require DYW2 for their function. This model can then be used to improve the predictions for uncharacterized E+ PPR proteins. The predictions are using a PPR code but also a list of potential binding sites. There are currently 795 identified editing sites corresponding to as many potential binding sites (Appendix A). Given the previous results, for E+ PPR proteins, this list can be reduced to the 272 editing sites depending on DYW2. Using the binding site predictions of Kobayashi et al. [17] for the 18 characterized E+ PPR proteins showed that filtering for DYW2-dependent sites strongly improved the ranks of their associated editing sites (Table 3). 

Appendix A provides the binding predictions extracted from Kobayashi et al. [17] for the 59 A. thaliana E+ PPR proteins to the DYW2 dependent sites. As at least two DYW2 dependent sites (M17869 and M234052) are not associated to E+ PPR proteins (Figure 1), it is not possible to reduce the list of candidate PPR proteins binding to DYW2 dependent sites. For the same reason, it is not possible to reduce the list of potential editing sites for non E+ PPR proteins to DYW2 independent sites. However, as the characterized DYW2-independent sites are so far exclusively associated to non E+ PPR proteins (Figure 1), it is possible to reduce the list of candidate PPR proteins binding to DYW2-independent sites by removing the E+ PPR proteins (Appendix A).

## 3. Discussion

A strong positive bias for sites associated to PPR-E+ proteins was recently identified in the RNAseq analysis of the *dyw2* mutant affected in a member of a small subfamily of PPR-DYW proteins [6]. Here, we strengthened this correlation by reanalyzing the *dyw2* dataset including 14 previously PPR-associated sites which were missed in the first analysis and by identifying the editing sites controlled by three new PPR proteins: OTP90, a DYW-type PPR and PGN and MEF37, and two E+-type PPR proteins. OTP90, PGN and MEF37 were shown to be mitochondrial editing PPR proteins involved in the editing of a total of 16 major editing sites.

### 3.1. OTP90, PGN and MEF37 Are Major Mitochondrial Editing Factors

*pgn* mutants were previously shown to be more sensitive to *Botrytis cinerea* infection [35]. Here, we identify PGN molecular function: this E+-type PPR is required for the editing of two mitochondrial sites, M8348 (*cox2_742*) and M165765 (*nad6_leader_-73*). The impacts of these editing sites on COX2 and NAD6 proteins as well as the activities of the mitochondrial complexes they belong, cytochrome c oxidase and complex I, respectively, were not characterized. Accordingly, it is not possible to speculate on the causality link between PGN molecular function and its involvement in pathogen sensitivity.

As PGN, MEF37 is a canonical E+-type PPR. While *mef37* mutants did not show any macroscopic phenotype, our RNAseq analysis shows that MEF37 is a major mitochondrial editing factor which is required for editing of 6 sites, M17869 (*ccmB_566*), M17884 (*ccmB_551*), M23217 (*rps3_1470*), M219378 (*mttb_387*), M308476 (*ccmC_179*) and M362343 (*atp4_138*), and which is also involved in editing of M362349 (*atp4_144*) and M215126 (*nad4_437*). Noticeably, two couples of these sites have very close positions (M362343/M362349 and M17869/M17884) questioning the binding site positions of MEF37 and the putative links between the corresponding editing defects (see below).

The mutation of OTP90, a DYW-type PPR, affects root growth at seedling stage and seed size. At the molecular scale, OTP90 is involved in the editing of 8 mitochondrial sites. Indeed, in *otp90*, editing is nearly abolished at 6 sites, M18355 (*ccmB_80*), M59321 (*nad1_500*), M219668 (*mttb_97*) and M17839 (*ccmB_596*), and strongly affected at M191687 (*ccmFc_1246*) and M308481 (*ccmC_184*). While the protein is dually targeted to mitochondria and plastids, we did not observe any defect in plastidial editing. We cannot exclude that OTP90 has another molecular function in these organelles. However, all DYW-type PPR identified so far are editing factors [26] with the exception of CRR2 which is involved in RNA cleavage in plastids [38] and our RNAseq analysis of *otp90* did not identified any defect in plastidial RNA. This absence of identified function of a PPR protein in a cell compartment where it is localized was already shown for OTP87 [39]. These silent editing factors could participate in the puzzling adaptability of the PPR family to organellar genome mutations especially at the RNA editing level.

### 3.2. OTP90, MEF37 and PGN Share Editing Sites with Several Other Editing Factors

Among the 16 major editing sites affected in *otp90*, *mef37* or *pgn* mutants, several were previously identified in other mutants. This is the case of all the editing sites affected in *mef37* or *pgn* mutants which are also differentially edited in *dyw2* (Table 1). This is also the case of the six major targets of OTP90 which are affected in *morf1* and *morf8/rip1* mutants [18,20]. 

Sharing a common editing site may reflect the need for the interaction of several editing factors to achieve editing. This was shown for CRR4 and DYW1 proteins which physically interact in order to edit plastidial ndhD_2 site [31] and, similarly, for DYW2 and E+-type PPR proteins such as CLB19 or SLO2 [6,32]. In this case, DYW1 and DYW2 proteins are proposed to bring a cytidine deaminase activity to the editosomes. 

MORF proteins have been reported to bind PLS class PPR proteins to enhance their affinity to target RNAs. Direct binding of MORF1 to the OTP90 *in vivo* and in yeast is consistent with the editing deficient phenotypes. Similarly, MORF1 may be required for increasing affinity between OTP90 and its target RNA sequence. It is unclear how MORF8 is involved in the RNA editing at the target sites of OTP90, since it did not bind to OTP90 at least in yeast. MORF1 may intermediate or support interaction between OTP90 and MORF8 as shown that MORF1 enhances interaction between MEF13 and MORF3 proteins [40].

In contrast, sharing a common editing site may also reflect a competition for site binding between different RNA binding proteins as well as indirect effects of mutations on binding sites of other RNA binding proteins. M17869 (*ccmB_566*) is affected in *mef37* as well as in *mef19*, a previously studied PPR mutant [33]. The PPR protein MEF19, encoded by the gene At3g05240, has 13 PPR repeats and is a member of the E-PPR proteins. In the *mef19* mutant line, RNA editing at M17869 (*ccmB_566*) was completely absent indicating that MEF19 is the single responsible specificity factor for this site and no other PPR protein can take its place for *cis*-element recognition [33]. The nucleotide at M17884 (*ccmB_551*), which is one of the targets of MEF37, is bound by the second PPR motif of MEF19 according to the PPR binding model. The two key amino acid residues for the PPR code, positions 6 and 1’ of the second PPR motif are an N and a D, respectively, which bind both U and C but preferentially to a U according to the PPR code. Therefore, transcripts of *ccmB* with an unedited C at position M17884 (*ccmB_551*) should be bound less efficiently by MEF19 than the edited version. These results strongly suggest that the RNA edition at M17884 (*ccmB_551*) is required for the binding of MEF19 and editing at M17869 (*ccmB_566*). Abolishment of an editing event leading to a lack of the downstream edition was also reported in *Physcomitrella patens*. While knock-out of PpPPR_71 gene completely lost RNA editing at *ccmFc-C122* [41], knock-down lines of PpPPR_65 reduced both *ccmFc-C103* and *ccmFc-C122* [42], suggesting editing at *ccmFc-C103* is required for the downstream site editing. Among the 795 putative *cis*-elements of editing sites in *Arabidopsis* organelles (pos -31 to -4 of known editing sites) 323 contain other editing sites which can influence the affinity of the PPR proteins as shown for the binding of MEF19 for the editing of M17869 (*ccmB_566*). Similarly, M362349 (*atp4_144*) is more edited in *mef37* while the edition of M362343 (*atp4_138*) which lies in the binding site of M362349 (*atp4_144*) is almost abolished. It suggests a higher affinity of the unknown PPR editing M362349 for the unedited M362343. Such secondary or even multiple sequential effects for downstream sites may be commonly present especially in mitochondria where many editing sites are closely located each other. This is somehow limiting reverse genetics based identification of PPR targets because in some cases it becomes impossible to separate the primary direct effect of a *ppr* mutant on its RNA target from secondary effects of sequence modification of the RNA target of another RNA binding protein.

This limitation can be exemplified by the study of PPME. PPME is so far one of the three non PLS-type PPR proteins involved in RNA editing. It is required for the editing of M234052 (*nad1_*898) and M234091 (*nad1_937*) but not M234082 (*nad1_*928) which stands between the 2 sites. In parallel, PPME was shown to bind M234052 (*nad1_*898) and M234082 (*nad1_*928) sites and not the downstream M234091 (*nad1_937*) site which is affected by its mutation [34]. Without further experiments, it remains unclear how this P-type PPR protein participates in the editing of mitochondrial transcripts.

### 3.3. Specificity of DYW2 for E+ PPR Proteins

All editing sites associated to E+ PPR so far are dependent on DYW2 for their edition but among the DYW2 dependent sites, M17869 (*ccmB_566*) is associated to MEF19, an E-PPR [33] and M234052 (*nad1_*898) is associated to PPME, a pure PPR [34]. 

As mentioned in the previous section, it is most likely that the impairment of M17869 (*ccmB_566*) edition in an E+ PPR mutant *mef37* and *dyw2* is a secondary effect of the lack of the direct target editing at M17884 (*ccmB_551*) which prevents the binding of MEF19. 

As also previously discussed, if PPME is required for the editing of M234052 (*nad1_*898) and M234091 (*nad1_937*), its molecular function in editing is not yet understood. In particular, the PPR code is not well conserved in this P-type PPR protein and it is unclear whether its function is to replace a canonical PLS-type protein or whether, like NUWA, it performs an unknown function in editing [34]. Moreover, only one of its two sites (M234052) is DYW2-dependent suggesting that DYW2 is not fundamentally required for the editing function of PPME. It is possible that M234052 (*nad1_*898) is edited by an unknown E+ PPR which requires both PPME and DYW2 to work. The best candidate PPR proteins for targeting this site according to the PPR code are two E+ PPR, At3g28640 and At3g28660 [17].

As a consequence the DYW2 criterion cannot be used to reduce the number of potential binding sites for other PLS-type PPR because it is not possible to consider that editing sites associated with non E+ PPR proteins are exclusively non DYW2-dependent.

On the other hand, the DYW 2 criterion is useful to limit the number of candidate targets of E+ PPR proteins. However, it needs to be noted that some E+ editing sites are not completely abolished in the *dyw2* mutant while they are in the corresponding E+ knock-out mutant. One explanation could be that the DYW2 protein was still present as traces in the *dyw2* mutant because of the use of the ABI3 promoter to bypass the embryo-lethality [6]. These traces could be sufficient to sustain the partial editing of a few sites. However, no *DYW2* transcript was detected by RNA-seq in these plants. It does not rule out the presence of traces of DYW2 but makes it unlikely. There are 6 DYW1-like proteins (DYW-PPR with very few PPR domains, no PG box and lack of conserved residues in the E2 domain): DYW1, DYW2, MEF8, MEF8s, AT2G34370 (DYW3) and AT1G29710 (DYW4). DYW1 is associated to CRR4 [31] but MEF8, MEF8s, DYW3 and/or DYW4 could be associated to some E+ PPR proteins. The editotype of the *mef8* mutant was characterized [43]. About a third of the MEF8-dependent sites are also DYW2-dependent and in these cases the editing is only partially abolished in either mutants or both (Appendix A).

### 3.4. Limitations of the Code for PLS-Type PPR Proteins

The current PPR codes are almost exclusively based on the known editing sites/PPR associations which allow deriving the binding preferences of each PPR motif [14,15,17,44]. This analysis relies on the assumptions that editing PPR proteins bind from -4 and upstream of the sites which edition is significantly reduced in the corresponding KO mutant, one PPR motif per base and that the RNA binding site is not edited. 

While the binding of PLS type PPR proteins to the nucleotides located in position -4 and upstream of the editing site, one base per PPR motif seems demonstrated by several studies [13,45], care should be taken when identifying PPR binding sites through the reduction of editing in the corresponding KO mutant. First, it was shown that the proper binding of PLS PPR to its target RNA does not always result in editing [46,47,48] meaning that this approach can miss true binding sites. Second, as previously discussed the reduction of editing in KO mutants can be indirect and thus could not correspond to *bona fide* binding sites. These indirect effects can be suspected when a reduction in editing is associated to a close-by variation of editing or splicing. Furthermore, the advent of NGS allows not only the detection of completely abolished editing sites but also simple decrease in editing. We chose an arbitrary threshold of 25% of editing decrease to consider a site as dependent of a particular protein. Should these sites be considered as putative binding sites? And if yes, should they be given the same weight as abolished editing sites in the estimation of the binding preferences of each PPR motif? These remain open questions. They could be solved by dedicated RNA-PPR binding experiments such as RNA electromobility shift assay (REMSA) or analysis of PPR footprints on RNAs. However, these methods remain challenging for editing PPR proteins which probably transiently interact with their targets.

Using the DYW2 criterion for the prediction E+-PPR binding implies that some potential sites are ignored even if they get very good prediction scores. A simple explanation is that current prediction systems are not accurate. However, factors other than PPR motifs can dictate the binding pattern of a PPR. For example, it was shown that RNA secondary structures can impede PPR binding *in vitro* [13,49,50] but at the same time, several PPR proteins are known to bind to sites predicted to form RNA hairpins [51,52] which suggests that other RNA binding proteins are necessary to make these sites available to PPR proteins. On the other hand, PPR proteins are probably competing with other RNA binding proteins (other PPR proteins, CRM proteins…) so that PPR proteins are probably not binding *in vivo* to all their potential binding sites.

## 4. Materials and Methods 

### 4.1. Plant Material, Phenotype Characterization and Complementation Assay

*Arabidopsis thaliana* ecotype Columbia-0 (Col-0) was used as wild type plant. The T-DNA insertional mutants were obtained from the ABRC stock center. T-DNA homozygous plants were selected by PCR genotyping using the primers described in Appendix A. The localization of the insertions was verified by sequencing of the PCR products. Plants were grown in chambers at 16 h/8 h light/dark cycles, 22 °C at day and 20 °C at night. For *in vitro* phenotype characterization of *otp90*, seeds were surface sterilized, sown in Murashige and Skoog (MS) solid medium with 0.8% (*W/V*) of agar-agar type E (Sigma Aldrich, Saint-Louis, MO, USA) and grown in a growth chamber (16-h light/8-h dark cycle, 22 °C, 50% hygrometry) after cold treatment for 48 h at 4 °C. Primary root length elongation was measured on 12 day old plantlets after cold treatment of seeds using EZ-Rhizo software [53]. 

For the complementation of the *otp90* mutant, a genomic 3595 bp fragment containing the promoter, the Open Reading Frame (ORF) and the terminator region of the *OTP90* gene was amplified using the primers described in Appendix A and cloned into the pGWB1 binary vector [54] by Gateway^TM^ (Invitrogen, Carlsbad, CA, USA). The *pgn* mutant was complemented with the ORF of *PGN* amplified with the primers described in Appendix A and cloned into the pGWB14 binary vector [55] by Gateway^TM^ (Invitrogen, Carlsbad, CA, USA). For the complementation of *mef37-2*, the ORF of the MEF37 gene was amplified using the primers described in Appendix A and cloned into the pMpGWB102 binary vector [56] by the In-Fusion cloning system (Takara Bio Europe, Saint-Germain-en-Laye, France). 

### 4.2. RNA Analysis

Total RNA was extracted from leaves using the NucleoZOL kit (Macherey-Nagel, Hœrdt, France) followed by a purification with the Agencourt RNACleanup XP beads (Beckmann-Coulter, Villepinte, France). The sequencing libraries were constructed using the TruSeq Stranded Total RNA with Ribo-Zero Plant (Illumina Inc., San Diego, CA, USA) and then sequenced with a Nextseq500 in single-end 75 bp. The RNA-seq data was analysed as described [57] except for the following modifications. The reads were mapped with STAR v2.7.0c [58] on the *A. thaliana* genome available from the release 39 of EnsemblGenomes but the mitochondrial genome was replaced by the Col-0 mitochondrial genome from the NCBI under the accession BK010421. The mapping parameters were --outSAMprimaryFlag AllBestScore --outFilterMultimapScoreRange 0--alignIntronMax 1.

### 4.3. Direct cDNA Sequence Analysis.

For direct cDNA sequencing, relevant cDNA fragments covering one or more editing sites were obtained by RT-PCR amplification according to published protocols [47]. At the RNA editing sites, cDNA sequences were evaluated for the respective C to T differences. RNA editing levels were estimated by the relative heights of the respective nucleotide peaks in the sequence analyses. Ratios between heights were calculated with the DNA Dynamo program. Sequences were obtained commercially from 4base lab (Reutlingen, Germany) or from Macrogen (Seoul, South Korea).

### 4.4. Subcellular Protein Localization

The OTP90 full-length ORF without stop codon was cloned into p0229-DsRed2 destination vector using Gateway cloning and the primers described in Appendix A. After transformation of A. thaliana Col-0 plants, Subcellular localization of the OTP90:DsRed2 fusion protein was observed in root and cotyledon cells of 10–15 days old primary transformants grown on MS solid medium using a spectral Leica SP2 AOBS confocal microscope (Leica Microsystems, Wetzlar, Germany) equipped with an argon laser and a HeNe laser. To confirm the mitochondrial localization, the seedlings were stained with 1 µM of Mito-tracker green marker (Invitrogen) for 15–30 min. Signals were detected using laser lines at 488 nm (Mito-tracker green; excitation/emission 488/510–530 nm) and 543 nm (DsRed2; excitation/emission 543/570–600 nm/chlorophyll autofluorescence; excitation/emission 543/600–700 nm). The images were coded green (Mitotraker green or chlorophyll autofluorescence) and red (DsRed2) giving yellow co-localization signals when green and red signals overlapped in merged images.

### 4.5. Yeast Two Hybrid Assays 

Sequences coding for OTP90 and the various MORF were cloned with the In-Fusion HD cloning system (Takara Bio Europe, Saint-Germain-en-Laye, France). OTP90 inserts were integrated into the binding domain containing vector pGBKT7, and MORF into the activator domain containing vector pGADT7 of the GAL4 Two Hybrid System 3 (Takara Bio Europe, Saint-Germain-en-Laye, France). The vectors were co-transfected for expression into yeast cells (PJ69-4A) according to the protocol. Yeast cells with both bait and prey vectors were cultured in synthetic dropout medium without Leu and Trp. 5 µL of suspended cells with an OD600 of 0.3 were dropped onto the various selection media plates. To detect strong interactions, 2.5 mM of 3-Amino-1,2,4-triazole was added in the media plates.

### 4.6. In Vivo Protein–Protein Interaction Assay with BiFC Analysis

For *in vivo* protein–protein interaction assays, OTP90, MORF1, MORF2, MORF3, MORF4, MORF5 and MORF9 were fused with either YFP-N (1-155) or YFP-C (156–239) at their C terminus, respectively and cloned under 35S promoter. OTP90 was cloned in pGWB1 binary vector [54] by LR recombination reaction (Invitrogen) whereas MORF proteins were cloned in pMDC123 vector [59] containing a multiple cloning site from pET41 (Merck Millipore Novagen®, Darmstadt, Germany) with In-Fusion HD cloning system (Clontech). Combinations of agrobacteria containing OTP90-YFP-N and each MORF-YFP-C, on one hand and OTP90-YFP-C and each MORF-YFPN, on the other hand, were used at equal concentration to agroinfiltrate 3-weeks old tobacco leaves. After 48 h of incubation at 21 °C, transformed tobacco cells were analyzed using a confocal microscope Leica, TCS SP5 II.

## Figures and Tables

**Figure 1 plants-09-00280-f001:**
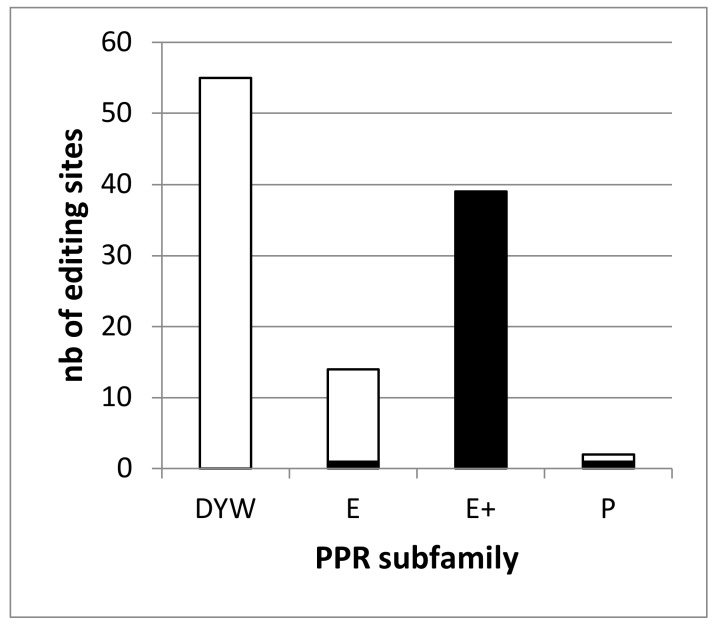
The editing defects in the *dyw2* mutant define two classes of RNA editing targets. The number of editing sites associated to a PPR of a given subfamily and depending on DYW2 (differentially edited in *dyw2-1* with a decrease of editing extent above 25%) is shown in black. The number of editing sites associated to a PPR of a given subfamily and independent of DYW2 is shown in white.

**Figure 2 plants-09-00280-f002:**
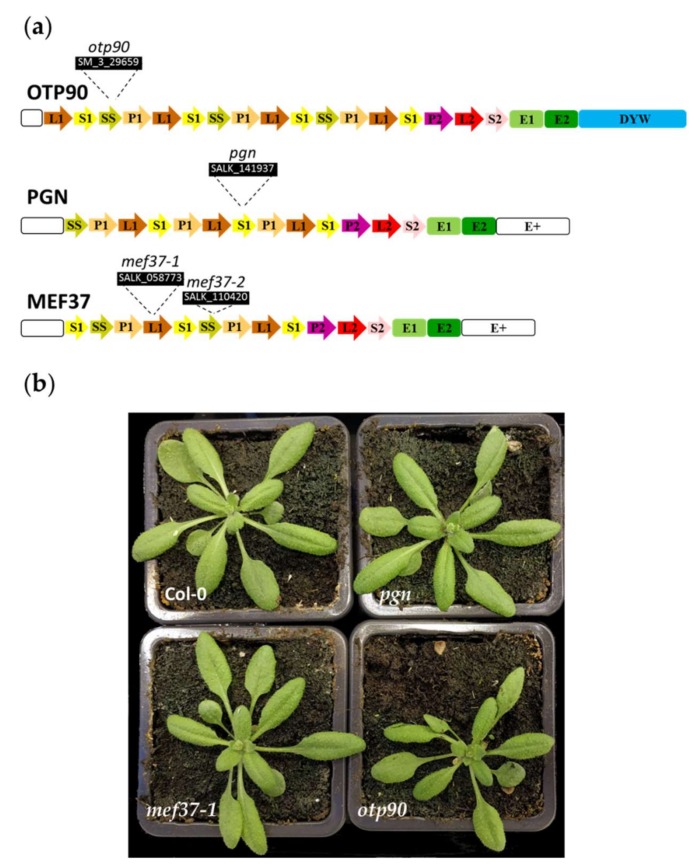
The three PPR proteins and their mutants. (**a**) Predicted domains of the proteins according to [37]. The positions of the T-DNA insertions in the corresponding genes are shown. (**b**) Phenotype of 25 day-old mutants compared to Col-0.

**Table 1 plants-09-00280-t001:** Differentially edited sites identified by RNA-seq in the *mef37-1*, *pgn* and *otp90* mutants compared to edit extent at these sites in *dyw2* mutant.

Mutant	Position ^1^	Site Name ^2^	WT ^3^	Mutant ^4^	ΔEE ^5^	Padj. ^6^	*dyw2* ^7^	ΔEE *dyw2* ^8^	Padj. *dyw2* ^9^
***mef37-1***	M17869	*ccmB_566*	90.5%	6.6%	−93%	0.000	13%	−86%	0.000
	M17884	*ccmB_551*	96.4%	1.4%	−99%	0.000	4%	−96%	0.000
	M23217	*rps3_1470*	72.4%	0.0%	−100%	0.000	1%	−99%	0.000
	M49473	*atp6_71*	0.6%	0.0%	−100%	0.002	ND	ND	ND
	M189896	*ccmFc_414*	3.5%	0.0%	−100%	0.002	0%	−98%	0.000
	M215126	*nad4_437*	98.1%	48.5%	−51%	0.000	71%	−28%	0.000
	M219378	*mttb_387*	42.8%	5.7%	−87%	0.000	1%	−96%	0.000
	M308476	*ccmC_179*	95.7%	0.9%	−99%	0.000	2%	−97%	0.000
	M362007	*nad4L_trailer_72*	90.2%	1.8%	−98%	0.000	2%	−97%	0.000
	M362343	*atp4_138*	92.1%	1.3%	−99%	0.000	5%	−95%	0.000
	M362349	*atp4_144*	7.7%	27.3%	253%	0.000	46%	331%	0.000
***pgn***	M8348	*cox2_742*	99.6%	0.0%	−100%	0.000	11%	−89%	0.000
	M165765	*nad6_leader_-73*	84.4%	0.2%	−100%	0.000	3%	−97%	0.000
***otp90***	M17839	*ccmB_596*	81.8%	23.0%	−72%	0.000	93%	11%	0.000
	M18355	*ccmB_80*	83.9%	5.9%	−93%	0.000	89%	12%	0.000
	M59321	*nad1_500*	73.9%	0.0%	−100%	0.000	87%	−3%	1.000
	M191687	*ccmFc_1246*	67.6%	12.7%	−81%	0,000	77%	26%	0.000
	M209816	*nad4_194*	9.6%	14.7%	54%	0.027	2%	−72%	0.000
	M209909	*nad4_111*	68.9%	84.6%	23%	0.000	92%	5%	0.000
	M219657	*mttb_108*	4.8%	0.2%	−95%	0.006	0%	−97%	0.000
	M219668	*mttb_97*	81.6%	1.9%	−98%	0.000	77%	−4%	1.000
	M233590	*matR_1950*	3.1%	7.8%	149%	0.049	0%	−94%	0.000
	M308481	*ccmC_184*	83.3%	5.6%	−93%	0.000	87%	4%	1.000
	M329728	*cox3_trailer_248*	0.6%	0.0%	−92%	0.000	2%	257%	0.000
	M362007	*nad4L_trailer_72*	86.9%	92.6%	7%	0.000	2%	−97%	0.000

^1^ M stands for “mitochondrion” followed by the position of the site in the mitochondrial genome of Col-0. ^2^ the name of the transcript followed by the position of the site after the start codon except when “trailer” or “leader” is mentioned in which case it is the position after the stop codon or before the start codon respectively. ^3^ editing extent in Col-0. ^4^ editing extent in the mutant. ^5^ variation of editing extent between the mutant and Col-0. ^6^ Pvalue adjusted with the Bonferroni correction for the mutant/Col-0 comparison. Values below 0.05 are considered significant. ^7^ editing extent in *dyw2*. ^8^ variation of editing extent between *dyw2* and Col-0. ^9^ Adjusted Pvalue for the *dyw2*/Col-0 comparison. ND: not detected. The values for the detected editing sites are given in Appendix A.

**Table 2 plants-09-00280-t002:** Mitochondrial C to U editing events analyzed in each mutant by direct cDNA sequencing. Editing levels are estimated for each gene in percentage related to the peak sizes (shown in Appendix A). Arabidopsis mitochondrial nucleotide editing events analyzed are indicated by their position in the transcript sequence.

Mutant	Position	Site Name	WT ^1^	Mutant ^2^	Compl ^3^
***mef37-2***	M17869	*ccmB_566*	100%	<5%	100%
	M17884	*ccmB_551*	100%	<5%	100%
	M23217	*rps3_1470*	85%	<5%	100%
	M215126	*nad* *4_437*	100%	70%	100%
	M219378	*mttb_387*	40%	<5%	50%
	M308476	*ccmC_179*	100%	<5%	100%
	M362343	*atp4_138*	100%	<5%	100%
	M362349	*atp4_144*	<5%	35%	<5%
***pgn***	M8348	*cox2_742*	85–90%	40–45%	85–90%
	M165765	*nad6_leader_-73*	85–90%	15%	85–90%
***otp90***	M17839	*ccmB_596*	100%	30-35%	100%
	M18355	*ccmB_80*	85–90%	<5%	75%
	M59321	*nad1_500*	100%	<5%	100%
	M191687	*ccmFc_1246*	55%	15%	90%
	M219668	*mttb_97*	80%	<5%	80–85%.
	M308481	*ccmC_184*	70%	<5%	85%

**^1^** Editing level in Col-0; **^2^** editing level in the mutant; **^3^** editing level in a corresponding complementation line. The symbol (-) means that no significant differences were found in mutant compared to Col-0.

**Table 3 plants-09-00280-t003:** Prediction of binding on their associated editing sites for the 18 characterized E+ PPR proteins using the system published in [17].

PPR	Position ^1^	Site Name ^2^	Rank ^3^	Rank DYW2 ^4^
AEF1	M26928	*nad5_1580*	1	1
AEF1	P12707	*atpF_92*	18	6
AHG11	M215187	*na* *d4_376*	1	1
CLB19	P69942	*clpP_559*	2	2
CLB19	P78691	*rpoA_200*	5	5
COD1	M6961	*cox2_698*	6	2
COD1	M6516	*cox2_253*	8	4
COD1	M209881	*nad4_1129*	34	14
CRR21	P116785	*ndhD_383*	1	1
CWM1	M235780	*nad5_598*	2	1
CWM1	M18007	*ccmB_428*	4	2
CWM1	M308760	*ccmC_463*	12	6
GRS1	M165940	*nad6_103*	1	1
GRS1	M361691	*nad4L_55*	2	2
GRS1	M160356	*rps4_377*	33	14
GRS1	M83057	*nad1_265*	109	41
MEF12	M235556	*nad5_374*	1	1
MEF13	M189532	*ccmFc_50*	1	1
MEF13	M189897	*ccmFc_415*	2	2
MEF13	M161857	*nad2_59*	3	3
MEF13	M215405	*nad4_158*	4	4
MEF13	M28242	*nad5_1916*	5	5
MEF13	M330460	*cox3_314*	6	6
MEF13	M27013	*nad5_1665*	47	21
MEF21	M330517	*cox3_257*	1	1
MEF25	M83014	*nad1_308*	12	7
MEF37	M215126	*nad4_437*	1	1
MEF37	M17884	*ccmB_551*	3	2
MEF37	M17869	*ccmB_566*	12	5
MEF37	M23217	*rps3_1470*	17	8
MEF37	M308476	*ccmC_179*	46	22
MEF37	M362343	*atp4_138*	57	24
MEF37	M219378	*mttB_387*	70	31
OTP72	M23724	*rpl16_440*	1	1
OTP80	P86055	*rpl23_89*	1	1
PGN	M8348	*cox2_742*	1	1
PGN	M165765	*nad6_leader_-73*	2	2
SLG1	M288290	*nad3_250*	1	1
SLO1	M215114	*nad4_449*	1	1
SLO1	M24992	*nad9_328*	2	2
SLO2	M241512	*nad7_739*	2	1
SLO2	M219621	*mttB_144*	4	2
SLO2	M361746	*nad4L_110*	76	23
SLO2	M219099	*mttB_666*	139	47
SLO2	M219620	*mttB_145*	177	65

^1^ M stands for “mitochondrion” followed by the position of the site in the mitochondrial genome of Col-0. P stands for “plastid”. ^2^ the name of the transcript followed by the position of the site after the start codon except when “leader” is mentioned in which case it is the position before the start codon. ^3^ rank of the corresponding editing site using the full list of potential binding sites. ^4^ rank using only the DYW2-dependent sites.

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
