# Peer review of "The Analysis of the Editing Defects in the *dyw2* Mutant Provides New Clues for the Prediction of RNA Targets of Arabidopsis E+-Class PPR Proteins"

_plants, 2020, doi:10.3390/plants9020280_

Round 1

Reviewer 1 Report

This manuscript by Malbert et al, entitled “The analysis of the editing defects in the dyw2 mutant provides new clues for the prediction of RNA targets of Arabidopsis E+-class PPR proteins”, describes detailed analyses of DYW2 and several PPR proteins, those involved in plant organellar RNA editing. The authors have focused the function of DYW2, which is known to be responsible for all RNA editing for E+-type PPR proteins. By incorporating the re-analyzed result of DYW2 editotype, the authors successfully improved the prediction score of RNA targets for E+-class PPR proteins.

 As described in this manuscript, discovery of the PPR code greatly improved our understanding about the selectivity of PPR-RNA binding, although the actual PPR-RNA interaction is not able to explain only by the PPR code. Therefore, other parameter(s) should be incorporated, including function of amino acid sequences of PPR protein other than those for PPR code, and other protein factors involved in RNA editing. This study could be a first example for a such attempt. Indeed, this manuscript, focusing DYW2, successfully demonstrated an improvement of the prediction score, in other word, in silico reconstitution of the site recognition of plant organellar RNA editing.

This reviewer suggests a few modifications, to improve readability of the manuscript or to fix the text appearance.

Minor

The sections 2.2 and 2.3 (Page 3, L114 and L120, respectively) are fragmented. The sections 2.2, 2.3, and 2.4 would be better to compile into a single section.

About the closing remark (P11, L386). The end of discussion, the authors pointed out the current limitations to elucidate PPR-RNA interaction and RNA editing in vivo. This reviewer would appreciate if the authors propose a possible next approach (or experiment) to improve our understanding of RNA editing, as a future prospect.

Page 9, L270: “(ref)” should be removed or put an appropriate reference.

Page 13, L455: “XXX” should be removed or fixed.

Author Response

The sections 2.2 and 2.3 (Page 3, L114 and L120, respectively) are fragmented. The sections 2.2, 2.3, and 2.4 would be better to compile into a single section.

 Response: we merged these sections

About the closing remark (P11, L386). The end of discussion, the authors pointed out the current limitations to elucidate PPR-RNA interaction and RNA editing in vivo. This reviewer would appreciate if the authors propose a possible next approach (or experiment) to improve our understanding of RNA editing, as a future prospect.

Response: 2 sentences have been added.

Page 9, L270: “(ref)” should be removed or put an appropriate reference.

 Response: thank you for pointing this out. The reference was added.

Page 13, L455: “XXX” should be removed or fixed.

Response: thank you for pointing this out. The accession number was added.

Reviewer 2 Report

This work from Malbert et al. tries to deepen the difficult issue of editosome in plants. In particular they use a bioinformatic pipeline to exploit the DYW domain criterion for the prediction of E+PPR binding sites.

The main topic is well argued and the criterion followed appears to be quite linear. Just a few comments:

Why didn't the authors test the performance of their prediction pipeline on public databases containing editing events (e.g. REDIdb or PED)? It looks very intriguing and could offer interesting ideas.

Is there a mathematical correlation between the minimum number of editing events and the sensitivity of their prediction algorithm? 

Would it not therefore be necessary to statistically review the arbitrary decrease in editing threshold (25%) within which a site is considered as a target?

Author Response

Why didn't the authors test the performance of their prediction pipeline on public databases containing editing events (e.g. REDIdb or PED)? It looks very intriguing and could offer interesting ideas.

Answer: We don’t really understand this suggestion because we don’t see how to test the performance of the pipeline using the data of REDIdb or PED. These databases aggregate and curate genome sequence databases to provide a catalog of RNA editing sites across many organisms. What we aimed at in this work is to show that current binding prediction pipelines can be improved mainly for E+ PPR when the list of DYW2 dependent sites is known. To test the performance of the pipeline outside A. thaliana, we need 2 pieces of information: a list of DYW2-dependent sites and a list E+-PPR dependent sites in the same organisms. Currently the list of DYW2-dependent sites is available only for Arabidopsis thaliana as it is the only organism in which the dyw2 KO mutant has been characterized. As far as we can tell, this list can’t be predicted based on the genome sequence so it doesn’t seem possible to get such lists for other organisms without the characterization of the corresponding dyw2 KO mutant. Similarly, we’re missing the lists of E+-PPR dependent sites.

  Is there a mathematical correlation between the minimum number of editing events and the sensitivity of their prediction algorithm? 

Would it not therefore be necessary to statistically review the arbitrary decrease in editing threshold (25%) within which a site is considered as a target?

Answer: We set an arbitrary threshold on the DEE to consider a site as DYW2-dependent. In Guillaumot et al. 2017, we set it to 10% following what was proposed in other publications (Bentolila et al, 2013; Diaz et al, 2017). In this study, we set it to 25% as the lowest DEE in dyw2 for an E+PPR associated editing site is 31.4% (cf site M18007 is dataS1, dyw2_editing) while the highest DEE in dyw2 for a non E+ PPR (except MEF19 and PPME) is  10.1%.

However, the first criterion to associate a site to DYW2 is the adjusted P-value of editing difference between Col-0 and the dyw2 KO mutant. This RNA-seq assay is screening the entire genome (not just a fixed list of known editing sites, cf material and methods). Given the Bonferroni correction, the more sites are tested the less sensitive the test becomes. However, despite this high stringency, we detected significant differences even for sites which are barely edited in both Col-0 and dyw2 (cf M234090) or for sites with a very small DEE (cf M209838). Are these differences biologically relevant? That’s why we (and others) are setting additional filters which may seem arbitrary.

Reviewer 3 Report

   Transcripts in both plant mitochondria and chloroplasts are subject to C-to-U editing, a post-transcriptional process in which specific C residues are targeted for deamination. Sites are targeted by the binding of pentatricopeptide repeat proteins (PPRs) in a sequence-specific manner. These proteins are divided into two classes, P-type PPR proteins and PLS-type PPR proteins. The latter class contains canonical, long and short variations of the PPR motif. PLS-PPR proteins are further categorized into subfamilies by their C terminal domains, which generally contain conserved E1, E2, E+, and DYW motifs. Available evidence strongly supports a model in which the catalytic activity is provided by the DYW domain, which shows homology to cytidine deaminases, but a subset of PLS-PPR proteins lacks the DYW domain. In these cases, the deamination activity is provided in trans by proteins in the DYW1-like class of proteins that contain a DYW domain, but lack certain other conserved elements. This manuscript examines the role of one such protein, DYW-2, by analyzing the effects of a dyw2 knockout mutant in detail and characterizing additional editing factor mutants. The authors then use these data to refine programs used to predict editing targets by E+ PPR proteins, which lack the DYW domain.

   Because this manuscript contains a reanalysis of existing dyw2 KO data, the initial impression was that it represents only an incremental advance. However, upon closer reading, the manuscript contains a significant amount of new data, including the identification of a large number of new differentially edited sites in dyw2 knockouts, convincing evidence of the functional relationship between DYW2 and E+ editing factors, and characterization of the targets of three additional editing factors, much of which is provided as supplementary data. This information is then used to improve prediction of editing sites that are likely targets of E+ type PPR proteins. The discussion, although fairly long, elaborates on the likely interdependence of certain editing sites and provides a thorough overview of the limitations of current algorithms and important outstanding questions.

Specific revisions:

The abstract requires editing to correct grammatical mistakes. Line 270: a reference is missing, please provide the (ref) Line 455: provide accession numbers currently annotated as (XXX) A detailed legend is needed for the Supplementary Data files. The legends should include a clear explanation of all column headings. Given the size of the error bars in Figure S4. The data do not seem to support the authors’ statement that the pgn mutant is more sensitive to Botrytis cinerea than Col-0.

Author Response

The abstract requires editing to correct grammatical mistakes.

Response : the abstract has been improved.

Line 270: a reference is missing, please provide the (ref)

Response: thank you for pointing this out. The reference was added.

Line 455: provide accession numbers currently annotated as (XXX)

Response: thank you for pointing this out. The accession number was added.

A detailed legend is needed for the Supplementary Data files. The legends should include a clear explanation of all column headings.

Response: thank you for pointing this out. The legends have been extended and added to each supplementary data file in the first sheet.

Given the size of the error bars in Figure S4. The data do not seem to support the authors’ statement that the pgn mutant is more sensitive to Botrytis cinerea than Col-0.

Response: We agree that the difference of sensitivity between Col-0 and pgn appears very small and variable but the statistical test does support it. And this difference was already published previously by another group.